# Comparison of Post-Thaw Motility and In Vitro Fertility Between Ejaculated and Epididymal Semen, and Seminal cfDNA Characterization in Pantaneiro Bulls

**DOI:** 10.3390/biology14050465

**Published:** 2025-04-25

**Authors:** Marcos Coura Carneiro, Alice Caroline Souza e Castro, Roberta Reis Silva, José Eduardo Vieira Chaves, Venâncio Augusto Oliveira Silva, Natalia Ernandes Capobianco, Paulo José Bastos Queiroz, Leonardo de França e Melo, Eleonora Araújo Barbosa, Margot Alves Nunes Dode, José Felipe Warmling Sprícigo

**Affiliations:** 1Escola de Veterinária e Zootecnia, Universidade Federal de Goiás (UFG), Goiânia 74690-900, Brazil; mcarneiro23@discente.ufg.br (M.C.C.); alicecaroline@discente.ufg.br (A.C.S.e.C.); robertars25@hotmail.com (R.R.S.); josechaves@discente.ufg.br (J.E.V.C.); venancioaugusto@discente.ufg.br (V.A.O.S.); pauloqueiroz@ufg.br (P.J.B.Q.); leonardo.franca@ufg.br (L.d.F.e.M.); 2Embrapa Recursos Genéticos e Biotecnologia, Laboratório de Reprodução Animal, Brasília 70770-917, Brazil; na.capobianco@hotmail.com (N.E.C.); margot.dode@embrapa.br (M.A.N.D.); 3Unidade Acadêmica de Serra Serra Talhada, Universidade Federal Rural de Pernambuco, Serra Talhada 56909-535, Brazil; eleonora.araujo@ufrpe.br

**Keywords:** indigenous cattle, semen cryopreservation, IVP, molecular marker

## Abstract

This study investigated the motility and fertility of frozen-thawed semen from Pantaneiro bulls, comparing samples collected via ejaculation and directly from the epididymis. The aim was to evaluate the potential of epididymal semen for use in conservation programs by assessing its post-thaw motility and in vitro fertility. Semen from five bulls was analyzed for fresh quality, cell-free DNA (cfDNA) levels, and sperm performance after freezing. Results showed that epididymal semen had fewer defects and exhibited superior motility after thawing compared to ejaculated semen. Despite these differences, both semen sources were equally effective in fertilizing oocytes in vitro. The characterization of cfDNA levels adds valuable insights to the field of male fertility and supports efforts to conserve indigenous cattle breeds. This research contributes to improving semen preservation techniques and highlights the motility of epididymal semen as a reproductive resource in livestock breeding and conservation strategies.

## 1. Introduction

The preservation of indigenous cattle is crucial to ensure the resilience and sustainability of the livestock system in the face of constantly evolving environmental and climatic challenges [1]. Regarding the Brazilian herd, the Pantaneiro breed originates from *Bos taurus* cattle of the Iberian Peninsula, which were brought to Brazil during colonization (~1600–1800). These animals have adapted to the Pantanal biome, characterized by high temperatures and large flooded areas during the rainy season [1]. Throughout more than 300 years of adaptation, cattle with increased robustness, prolificacy, and resilience to the region’s conditions have been naturally selected [2,3,4]. However, since the beginning of the 20th century, the Nellore breed has been replacing local breeds due to its higher production [2,4]. This replacement has resulted in losses of genetic diversity, compromising access to genes and genetic combinations that could be beneficial in future livestock breeding endeavors [4].

Given this scenario, the utilization of reproductive technologies becomes crucial for preserving the genetic material of local breeds, enabling its incorporation into breeding programs and facilitating the dissemination of these genes. Semen cryopreservation is a commonly employed method and is an essential procedure for the management of germplasm banks, thereby contributing to biodiversity conservation, safeguarding of endangered species, and the continuation of reproduction in valuable animals that are already deceased [5,6]. Furthermore, this technique allows for long-term storage, genetic dissemination of superior animals across generations, long-distance semen transport, and insemination of females at the optimal time, eliminating the need for a breeding male on the farm [6,7].

Despite these benefits, cryopreservation can cause physical and functional damage to sperm, resulting in decreased fertility [8]. Moreover, bulls often exhibit significant individual variation in sperm quality post-cryopreservation, which affects their capacity for in vitro embryo production [9]. Hence, identifying semen markers pre-cryopreservation that can predict freezability and subsequent in vitro embryo production offers significant advantages [10]. Recent studies have demonstrated that the presence of cell-free DNA (cfDNA) in seminal plasma is a potential marker of sperm quality in humans [11] and Nellore cattle [10]. cfDNA in seminal plasma consists of small DNA fragments released during cellular processes such as apoptosis or necrosis. This molecule has been identified as a potential biomarker for sperm quality and fertility, as elevated cfDNA levels may indicate cellular stress or damage. In humans [11] and livestock [10], cfDNA has been correlated with sperm membrane integrity, capacitation status, and fertilization outcomes [10], making it a promising non-invasive marker for male fertility assessment [10,11]. However, information is limited regarding cfDNA in bovine semen, particularly in bulls of Brazilian indigenous breeds such as the Pantaneiro.

The successful cryopreservation of bovine semen collected from ejaculates opens the possibility of using epididymal sperm cells, which can be retrieved post-mortem, supporting reproductive programs and genetic preservation efforts, particularly for indigenous cattle. Cryopreserving epididymal sperm has been studied in various domestic [12,13,14,15] and wild species [16,17,18]. This technique is crucial in genetic conservation programs, as it enables the preservation of genetic material, even after death [19]. After preservation, this genetic material can be utilized in artificial insemination and in vitro embryo production, thereby conserving valuable genetic diversity [20,21]. Furthermore, some research has shown that epididymal sperm are more resistant to freezing than ejaculated sperm [22,23]. Cunha et al. [24] further demonstrated that treating epididymal sperm with heparin accelerates fertilization, enabling a reduction in co-incubation time and thus improving in vitro fertilization outcomes for bovine spermatozoa. To date, no studies have evaluated the post-thaw motility and quality of epididymal sperm from Pantaneiro breed bulls, nor has the potential of these sperm cells to support in vitro embryo production been explored for this indigenous breed. Given this gap in the literature, our study not only investigates semen motility but also examines its ability to support embryo development through in vitro fertilization.

Considering the potential of cfDNA as a predictor of in vitro fertility, this study aimed to quantify the cfDNA content in the seminal plasma of Pantaneiro bulls. Additionally, we evaluated the motility and quality of semen from these bulls before and after thawing in samples obtained from both the epididymis and ejaculate. Crucially, we also assessed the in vitro embryo development following fertilization with both types of semen, providing new insights into their reproductive potential.

## 2. Materials and Methods

Unless otherwise indicated, all reagents and chemicals used in this study were purchased from Sigma-Aldrich (St. Louis, MO, USA).

### 2.1. Animals Selected for the Study

All animal procedures were performed in accordance with Brazilian Law for Animal Protection and were approved by the Ethics Committee for the Use of Animals at the Federal University of Goiás (Case No. 078/2021).

Five Pantaneiro bulls, aged between 24 and 48 months and raised under identical management conditions, were utilized for the study. The animals were housed at the EVZ/UFG facilities in Goiânia-GO. They were kept in three paddocks and provided with ad libitum water, Tifton 85 hay (4 kg/animal/day), and a concentrate (7 kg/animal/day) composed of 10% soybean meal, 88% corn, 1% urea, and 1% mineral supplement. The animals were evaluated for weight (438 ± 9.9 kg) and body condition score (average, ~3.6). Additionally, they underwent clinical and andrological examinations, and only those meeting the health and andrological criteria were selected for the study.

### 2.2. Experimental Design

The five bulls underwent two seminal collections, separated by a fifteen-day interval. The first collection was performed via electroejaculation, with seminal samples used for both the extraction and quantification of plasma cfDNA, and for cryopreservation followed by post-thaw evaluation. The second collection involved post-mortem retrieval of samples through epididymis sectioning and puncture, which were subsequently cryopreserved and evaluated post-thaw.

The quality of fresh semen from both origins (ejaculate and epididymis) was assessed before the cryopreservation procedure (n = 5 bulls). Post-thaw, seminal samples from the five bulls, representing both origins (n = 3 straws/bull), were evaluated for sperm cell kinetics using computer-assisted sperm analysis (CASA) at two time points: immediately after thawing and after six hours of in vitro incubation.

Finally, an experiment was conducted with additional cryopreserved straws (n = 8/bull/origin) to assess the in vitro embryo development potential of samples collected from either ejaculate or epididymis.

#### 2.2.1. Collection of Ejaculated Semen and Acquisition of Seminal Plasma

Semen was collected using the electroejaculation method (Eletrogen, SA 200, Presidente Prudente, Brazil). Intermittent electrical stimuli ranging from 200 to 500 mA with intervals between 2 s and 3 s were applied to ensure proper animal stimulation, with some bulls experiencing stimuli lasting up to 5 s. After ejaculation, half of the volume was used for cryopreservation purposes. The remaining half was used for plasma isolation, performed through two consecutive centrifugations at 4 °C. The first centrifugation was at 400× *g* for 10 min to avoid cell lysis, followed by a second at 16,000× *g* for 5 min (adapted from Ponti et al. [25]) for total cell separation. After the centrifugation steps, 0.5 mL aliquots of seminal plasma were frozen at −80 °C for subsequent DNA extraction.

#### 2.2.2. Collection of Epididymal Spermatozoa

Fifteen days after the initial semen collection via ejaculate, the five bulls were slaughtered at a commercial slaughterhouse. Immediately afterward, their testicles and epididymides were transported to the laboratory within 2 h. Upon arrival at the Animal Reproduction Laboratory of the Federal University of Goiás, the testicles and epididymides were cleaned with a saline solution (0.9% NaCl) and 70% alcohol. Spermatozoa were collected from the epididymal tail by making several incisions and applying pressure (manually applied by the technician during testicular manipulation) to extrude the fluid [10].

#### 2.2.3. Fresh Seminal Analysis

A 10 µL aliquot was withdrawn, diluted in Ringer-lactate solution, and used for seminal quality evaluations of fresh semen under light microscopy (Olympus BX41 microscope, Hachioji, Japan) by three evaluators, assessing parameters of vigor, motility, and concentration according to the Manual for Andrological Examination and Semen Evaluation in Animals [26]. Additionally, sperm morphology analysis was conducted using phase-contrast microscopy; 200 cells were counted per slide. The morphological abnormalities were recorded as percentages of minor, major, and total defects [26].

#### 2.2.4. Semen Freezing and Thawing

After seminal analysis, sperm from ejaculate or epididymal samples were diluted using Optidux^®^ (Reprodux, Campinas, SP, Brazil), a protein-free semen diluent devoid of animal-derived components, composed of phospholipids (liposomes), carbohydrates, mineral salts, buffering agents, antioxidants, glycerol, and antibiotics (gentamicin, tylosin, lincomycin, and spectinomycin), loaded into 0.25 mL straws, and cooled to 4 °C for 4 h inside a refrigerator (Electrolux, Stockholm, Sweden), maintaining a stabilized temperature. Subsequently, the straws were placed in liquid nitrogen vapor (approximately −20 °C) for 20 min and then immersed in liquid nitrogen at −196 °C [23].

For semen thawing, the method recommended by the Brazilian College of Animal Reproduction [26] was employed, which involved direct immersion of the straw in water at a temperature between 35 °C and 37 °C for 30 s.

#### 2.2.5. Evaluation of Post-Thaw Semen Motility

For the evaluation of sperm motility after thawing, three straws per bull per origin (ejaculate and epididymis) were used. The first sample was acquired immediately after thawing at 0 h. Additionally, the frozen/thawed semen was incubated in an IVF medium consisting of Tyrode’s Albumin Lactate and Pyruvate (TALP) supplemented with 0.5 mM penicillamine, 0.25 mM hypotaurine, 25 µM epinephrine, and 10 µg/mL heparin under in vitro conditions at 38.5 °C and 5% CO_2_ for 6 h. At this point, a second sample was collected for analysis. Samples at both time points underwent a motility assay assessed using the computer-assisted sperm analysis (CASA) system IVOS 12.3 (Hamilton-Thorne Biosciences^®^, Beverly, MA, USA) with a pre-configured setup according to the manufacturer’s manual for bovine sperm analysis. For CASA analysis, 10 µL of semen was placed in a preheated reading chamber (Makler, Santa Ana, CA, USA). At least five fields were manually selected for reading and analysis. Measured parameters included total and progressive motility, curvilinear velocity (VCL), average path velocity (VAP), progressive linear velocity (VSL), straightness (STR), and linearity (LIN). In addition, sperm morphology analysis was performed using phase-contrast microscopy, with 200 cells counted per slide [27,28,29].

#### 2.2.6. Isolation and Quantification of cfDNA

Total cfDNA was isolated using the Quick-cfDNA/cfRNA Serum and Plasma kit (Zymo Research, Irvine, CA, USA) following the manufacturer’s recommendations. For each bull, three independent total cfDNA samples were isolated from 1 mL of seminal plasma. The total cfDNA was quantified using a NanoDrop^®^ spectrophotometer (Thermo Fisher Scientific, Waltham, MA, USA), and the average of three replicates was used.

#### 2.2.7. In Vitro Embryo Production (IVP)

For in vitro embryo production, ovaries from crossbred cows (*Bos indicus* × *Bos taurus*) were collected at a local slaughterhouse and transported to the laboratory. Follicles ranging from 3 to 8 mm were aspirated, and good-quality cumulus-oocyte complexes (COCs) were selected. The oocytes were matured in a maturation medium composed of TCM-199 supplemented with 10% fetal bovine serum (FBS), FSH, L-glutamine, and antibiotics, for a period of 22 to 24 h in an incubator at 38.5 °C with 5% CO_2_. After maturation, the oocytes were transferred to drops containing IVF medium supplemented with penicillin, hypotaurine, epinephrine, and heparin. Subsequently, they were co-incubated with spermatozoa from each bull, the semen dose was thawed at 36 °C for 30 s, and then viable spermatozoa were selected by centrifugation in a discontinuous Percoll gradient (GE^®^ Healthcare, Piscataway, NJ, USA) using 400 μL of 90% Percoll and 400 μL of 45% Percoll in a 1.5 mL microtube, at 700 g for 5 min. After 18 h of co-culture, the structures were transferred to in vitro culture in synthetic oviductal fluid (SOF) medium supplemented with myo-inositol and 5% FBS [23], then cultured at 38.5 °C in an incubator with 5% CO_2_ for eight days. After IVF, cleavage on D2 and blastocyst rates were evaluated on D6, D7, and D8 [30].

#### 2.2.8. Statistical Analysis

Data were analyzed using PROC GLIMMIX in SAS Studio^®^ OnDemand for Academics 2022 (SAS 2022, Cary, NC, USA). A mixed-model ANOVA was employed to account for fixed effects (group, time, interaction) and random effects (replication). For post-hoc comparisons, Tukey’s HSD test was used to adjust for multiple comparisons, with significance set at *p* ≤ 0.05. Residuals were checked for normality and homogeneity, and log-transformed where necessary.

## 3. Results

To quantify cell-free DNA (cfDNA) in seminal plasma, samples were collected from the ejaculate of each bull. Figure 1 displays the cfDNA concentrations obtained from the evaluated samples. The cfDNA concentrations ranged from 11.4 ng/µL to 50.9 ng/µL, with a median concentration of 31.1 ng/µL.

Regarding the cellular portion of the seminal samples, the fresh semen results from two experimental groups (Ejaculate vs. Epididymis) are presented in Table 1. Briefly, semen from Ejaculate exhibited significantly higher (*p* ≤ 0.05) volume, as well as both minor and total sperm defects compared to Epididymis. Among minor defects, the results revealed a decrease in the occurrence of distal cytoplasmic droplets (10.7% vs. 25.4%) and an increase in the presence of coiled tails (32.3% vs. 12.0%) in samples obtained from Ejaculate and Epididymis, respectively. The sperm concentration was higher (*p* ≤ 0.05) in Epididymis samples. No significant differences (*p* > 0.05) were observed between the groups for motility, vigor, membrane motility, or major defects (Table 1).

The evaluation of post-cryopreservation seminal quality was conducted using computerized sperm motility analysis (CASA) at two distinct time intervals: immediately post-thawing (0 h) and after a six-hour (6 h) incubation period in IVF media within controlled in vitro conditions (Figure 2). Notably, among the array of variables scrutinized, only VLS exhibited no statistically significant disparity (*p* > 0.05) between the two groups immediately following thawing (0 h). Noteworthy findings unveiled that samples obtained from the epididymis at 0 h showcased a markedly higher percentage of total and progressive motility, VAP, VCL, STR, and LIN percentages (*p* ≤ 0.05). Upon the second assessment (6 h), metrics for VAP, VCL, STR, and LIN displayed no significant differences (*p* > 0.05) between the Epididymis and Ejaculate sperm cell groups. However, total and progressive motility and VSL were notably elevated in epididymis-derived sperm samples (*p* ≤ 0.05). The analysis of the experimental group’s patterns over time revealed that only VSL in the Epididymis samples and VCL in the Ejaculate samples exhibited no significant variance (*p* > 0.05) across the two designated time intervals.

Finally, we evaluated the in vitro embryo development after in vitro fertilization using spermatozoa from ejaculate and epididymis (Table 2). For this analysis, semen from two bulls, whether from ejaculate or epididymis, was excluded due to suboptimal post-thaw sperm quality (Appendix A), as defined by the criteria outlined by the Brazilian College of Animal Reproduction [26]. The objective of the study was to evaluate embryonic development from oocytes fertilized in vitro with spermatozoa collected from either the ejaculate or epididymis. Despite some differences in fresh quality or post-thaw motility, no significant differences (*p* > 0.05) were observed in cleavage rates on D2 or in embryonic development on D6, D7, or D8.

## 4. Discussion

In conservation biology and sustainable livestock farming, preserving indigenous cattle breeds is essential for maintaining genetic diversity and resilience to environmental changes. These breeds offer unique genetic traits suited to specific regions, supporting biodiversity and more sustainable farming practices [31]. Reproductive technologies like semen cryopreservation and in vitro embryo production play a key role in conserving these genetic resources, especially for endangered breeds, allowing the use of sperm from deceased bulls for artificial insemination [32] and embryo production [24]. This helps prevent the loss of genetic material and supports the continuation of diverse cattle populations.

The present study aimed to evaluate fresh semen quality and, more importantly, the post-thaw motility and in vitro fertility potential of semen obtained from both ejaculated and epididymal sources in Pantaneiro bulls. Initial assessments of fresh semen revealed differences in sperm quality between the two sources, particularly in the presence of morphological defects. The quantity of defects displayed by ejaculated spermatozoa is considered acceptable for this parameter and species [33]. Moreover, the higher volume observed at the moment of ejaculation is due to the addition of seminal plasma to the cellular portion by the accessory glands, which leads to an increase in the volume of ejaculate compared to semen collected from the epididymis [29].

Despite the analysis of fresh semen, the primary focus of this study was on post-cryopreservation performance, where epididymal spermatozoa exhibited higher motility and fertility compared to ejaculated sperm. During cryopreservation and thawing, oxidative stress can cause damage to spermatozoa, leading to decreased motility and potentially hindering fertilization [34]. Furthermore, the sperm freezing process can increase their susceptibility to the actions of free radicals, which can damage the sperm membrane [35], reducing motility and quality [24]. This study demonstrates better sperm quality after thawing epididymal spermatozoa, suggesting that the protein profile and the presence of antioxidants in the epididymal fluid could be responsible for this increase in resistance to the freezing process. Additionally, the study aimed to characterize the concentration of cfDNA in seminal plasma, which may serve as a potential biomarker for semen quality and fertility.

The findings of this study, where epididymal sperm showed superior motility and kinetic parameters post-incubation compared to ejaculated sperm, are consistent with research highlighting the better post-thaw resilience of epididymal sperm. This is supported by studies in humans [36] and equines [37], which have shown that epididymal sperm maintains higher motility and motility after cryopreservation. The superior post-thaw performance of epididymal sperm is often attributed to its isolation from seminal plasma, which reduces the risk of premature capacitation and membrane destabilization, factors that can compromise sperm integrity during the freeze-thaw cycle [38]. Furthermore, Moore et al. [39] noted that seminal plasma proteins in ejaculated sperm could negatively impact cryopreservation outcomes, further explaining the enhanced resistance of epididymal sperm to cryoinjury. Similarly, Cunha et al. [24] observed that heparin treatment of epididymal sperm accelerated fertilization rates and allowed for reduced co-incubation times during IVF, further highlighting the advantages of using epididymal spermatozoa for assisted reproductive technologies (ART). The distinct regulatory environment of the epididymis, coupled with the enhanced performance of epididymal sperm under certain treatments like heparin, underscores the potential benefits of using epididymal sperm in reproductive interventions. Together, these findings emphasize that epididymal sperm may offer distinct advantages in ART, particularly in terms of motility and motility [24].

Regarding in vitro fertility, as depicted in Appendix A, two bulls out of the five initially selected were excluded from the in vitro embryo production experiment due to low post-thaw motility in the epididymis group. To uphold the integrity of the study and minimize potential biases, these bulls were subsequently eliminated from both experimental groups. The embryo development was similar for both experimental groups. During the process of in vitro fertilization (IVF), the contents of the semen straw are subjected to Percoll^®^ gradient sperm selection, which isolates a population of sperm with enhanced motility. This technique segregates viable sperm from non-viable ones, ensuring a purified sperm population for subsequent procedures in the IVF process [29]. Hence, during the fertilization process, a significant proportion of spermatozoa brought into contact with cumulus-oocyte complexes (COCs) would exhibit robust motility, indicating a comparable fertilization potential among this sperm population. This outcome aligns with previous studies, demonstrating consistent results in embryo production irrespective of whether sperm from the epididymis or ejaculate was utilized [40,41].

Although IVF techniques have been extensively studied, few articles compare the fertilization potential of epididymal sperm to ejaculated sperm. Martins et al. [33] found that some individuals produce fewer embryos using epididymal sperm, but they still emphasized its potential in IVF procedures. In our study, a slight tendency toward a reduced number of Day 7 embryos was observed following IVF with epididymal sperm; however, no statistically significant difference was detected. It is possible that, with a larger sample size and continued analysis, this difference may become statistically evident. Cunha et al. [24] observed that epididymal sperm treated with heparin produced embryo development rates of approximately 54%, comparable to those obtained with ejaculated sperm. The study also highlighted that the use of heparin significantly reduced co-incubation times, enhancing the overall efficiency of the IVF process. This high rate of embryo production underscores the potential of epididymal sperm in ART, particularly in situations where ejaculated sperm may not be available or viable. The findings suggest that with proper treatment, such as heparin addition, epididymal sperm can be as effective as ejaculated sperm for generating embryos, making it a valuable resource in reproductive technologies.

Shifting from the comparison of sperm sources, it is also important to consider other factors influencing fertility, such as the concentration of cfDNA in seminal plasma. In the present study, the cfDNA in the seminal plasma ranged from 11.4 ng/µL to 50.9 ng/µL. Recently, Dode et al. [10] demonstrated a variation from 15.23 ng/µL to 519.71 ng/µL in the ejaculated plasma of nine Nellore (Bos indicus) bulls. Moreover, the authors found a positive correlation between cfDNA concentration and in vitro embryo development potential. Despite some of this data, cfDNA may be associated with other cellular events, most of them linked to neoplasia. In humans, discrepant concentrations were also observed when evaluating seminal plasma from patients with prostatic neoplasia, ranging from 508 ng/µL to 4800 ng/µL [25], and from healthy men, ranging from 52.9 ng/µL to 62.5 ng/µL [42]. Thus, in humans, a correlation was found between the concentration of free cfDNA in seminal plasma and certain pathologies.

In reproductive biology, there is growing evidence supporting the association between cfDNA levels in seminal plasma and sperm fertility potential. Dode et al. [10] categorized bulls based on the concentration of cfDNA in their seminal plasma and found that those with lower cfDNA levels displayed greater sperm membrane stability. According to these authors, membrane destabilization is a precursor to sperm capacitation, and premature destabilization can lead to cell death and loss of fertilization potential. Therefore, higher membrane stability in bulls with lower cfDNA concentrations may indicate better overall semen quality. Interestingly, the same study found that while lower cfDNA levels were beneficial for sperm membrane stability, bulls with higher cfDNA concentrations produced more embryos in vitro, suggesting a potential role for cfDNA as a marker of both embryonic development and fertility [10].

In the current study, due to the limited number of Pantaneiro bulls, it was not possible to establish a statistical correlation between cfDNA concentration and fertility or motility. However, the two bulls that performed well in the CASA assay exhibited lower cfDNA levels in their seminal plasma compared to other bulls. Although this observation does not reach statistical significance, it highlights the need for further investigation to better understand the role of cfDNA as a potential biomarker for sperm fertility and motility, as suggested in earlier studies on reproductive and cancer biology.

## 5. Conclusions

This study demonstrates the superior post-thaw motility of epididymal spermatozoa compared to ejaculated sperm in Pantaneiro bulls, along with similar blastocyst production rates following IVF. Additionally, the concentration of cfDNA in the semen of these animals was characterized. These findings contribute to the conservation of endangered cattle breeds and advance basic research on in vitro fertility, supporting the development of effective reproductive biotechnologies.

## Figures and Tables

**Figure 1 biology-14-00465-f001:**
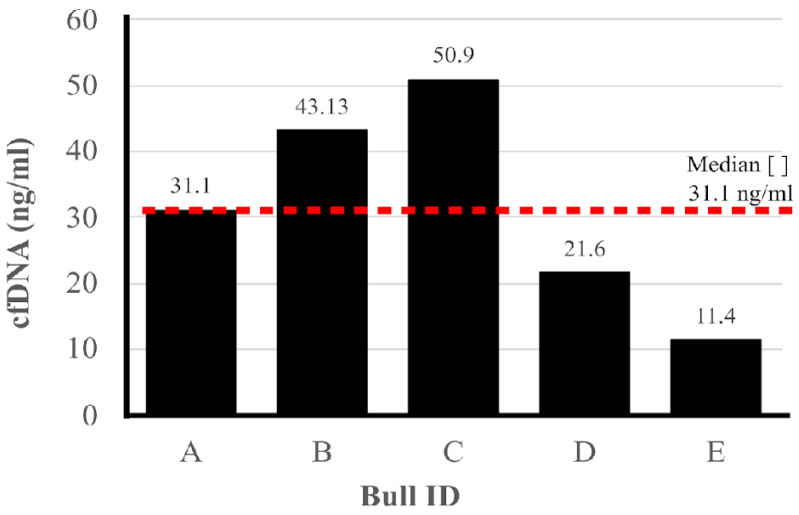
Quantification of concentration cell-free DNA (cfDNA) in seminal plasma. On the *X*-axis, A, B, C, D, and E are the identification of each bull. On the *Y*-axis, the cfDNA concentration in seminal plasma (ng/µL). The dashed line represents the median cfDNA concentration obtained from the five Pantaneiro bulls.

**Figure 2 biology-14-00465-f002:**
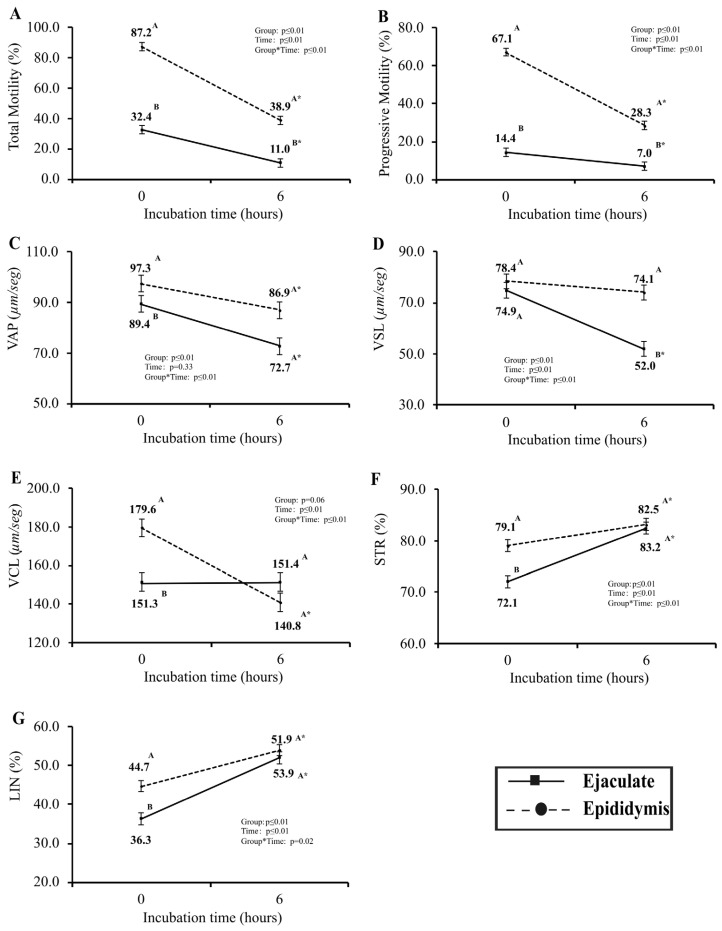
Total motility (**A**), progressive motility (**B**), VAP (**C**), VSL (**D**), VCL (**E**), STR (**F**), and LIN (**G**) were evaluated at 0 h and at 6 h of in vitro incubation post-thawing on Epididymis (dashed line) and Ejaculate (solid line). ^AB^ Different letters indicate differences between Epididymis and Ejaculate groups at specific time points, and the asterisk (*) indicates differences over time within the same experimental group, according to PROC GLIMMIX (*p* ≤ 0.05) on SAS.

**Table 1 biology-14-00465-t001:** Characteristics of sperm cells evaluated in fresh semen of Pantaneiro bulls collected from Ejaculate or Epididymis (mean ± standard error).

Sample Parameter	Ejaculate	Epididymis	*p* Value *
Bulls N	5	5	n/a
Volume (mL)	9.4 ± 2.0	3.4 ± 0.5	<0.01
Concentration (sptz × 10^6^)	480 ± 124.0	716 ± 147	<0.01
Total motility (%)	83.2 ± 3.6	87.0 ± 2.4	0.81
Vigor (1–5)	4.0 ± 0.4	4.4 ± 0.5	0.90
Plasma membrane integrity (%)	82.1 ± 6.3	83.5 ± 3.6	0.79
Minor defects (%)	15.8 ± 3.8	1.8 ± 1.8	<0.01
Major defects (%)	7.1 ± 1.6	6.1 ± 0.8	0.61
Total defects (%)	22.8 ± 4.2	7.8 ± 1.0	<0.01

n/a = non analyzed. * Data analyzed by PROC GLIMMIX on SAS, statistical significance was assumed at *p* ≤ 0.05.

**Table 2 biology-14-00465-t002:** Number of bulls, replicates, and oocytes (N) and number (n) and percentage (%) of cleavage on D2 and blastocyst on D6, D7, and D8 of development in groups fertilized with sperm cells collected from Ejaculate or Epididymis of Pantaneiro bulls.

Variable	Group		*p* Value *
	Ejaculate	Epididymis	
Bulls N	3	3	n/a
Replicates N	8	8	n/a
Oocytes N	525	500	n/a
Cleavage D2, % (n)	49.6 (261)	44.2 (221)	0.61
Blastocyst D6, % (n)	24.4 (128)	21.8 (109)	0.71
Blastocyst D7, % (n)	26.1 (137)	22.2 (111)	0.74
Blastocyst D8, % (n)	26.3 (138)	22.4 (112)	0.76

n/a = non analyzed. * Data analyzed by PROC GLIMMIX on SAS, statistical significance was assumed at *p* ≤ 0.05.

## Data Availability

The data presented in this study are available on request from the corresponding author.

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
