# Peer review of "Comparison of Post-Thaw Motility and In Vitro Fertility Between Ejaculated and Epididymal Semen, and Seminal cfDNA Characterization in Pantaneiro Bulls"

_biology, 2025, doi:10.3390/biology14050465_

Round 1
Reviewer 1 Report
Comments and Suggestions for Authors
The authors investigated the motility and fertility of frozen and thawed sperm from ejaculated and epididymal semen in Pantaneiro bull, and found that epididymal semen had fewer defects and showed better motility after thawing than ejaculated semen.
These results would suggest alternative way of sperm cryopreservation, however, there are some points to be clarify in the manuscript.
General points:
Why does using epididymal sperm other than ejaculated sperm enhance reproductive programs and genetic preservation (Line 77-78, 318-319)? Using ejaculated semen should be more advantageous for reproductive programs and genetic preservation efforts, since it can be obtained repeatedly without sacrificing bulls. Additionally, by calculating data in Table 1, both numbers of motility and non-defect sperm were higher in ejaculated than epididymal semen. Moreover, embryonic development from oocytes fertilized in vitro with spermatozoa collected from the ejaculate were superior comparing to that from the epididymis, even though no statistical difference was detected (Table 2).
It is unclear what the authors suggest the significance of cfDNA is in the present study.
Specific points:
Line 111: Standard error (deviation) should be indicated.
Line 145: How did it apply the pressure?
Line 218: Is 31.1 ng/ml average of median (Figure 1)? What does “[]” mean?
Line 259: Was there a relation between suboptimal post-thaw sperm quality and cfDNA (cannot access Sup. File 1). Which specific individuals were excluded?
Author Response
The authors investigated the motility and fertility of frozen and thawed sperm from ejaculated and epididymal semen in Pantaneiro bull, and found that epididymal semen had fewer defects and showed better motility after thawing than ejaculated semen.
These results would suggest alternative way of sperm cryopreservation, however, there are some points to be clarify in the manuscript.
General points:
Comments 1. Why does using epididymal sperm other than ejaculated sperm enhance reproductive programs and genetic preservation (Line 77-78, 318-319)? Using ejaculated semen should be more advantageous for reproductive programs and genetic preservation efforts, since it can be obtained repeatedly without sacrificing bulls. Additionally, by calculating data in Table 1, both numbers of motility and non-defect sperm were higher in ejaculated than epididymal semen. Moreover, embryonic development from oocytes fertilized in vitro with spermatozoa collected from the ejaculate were superior comparing to that from the epididymis, even though no statistical difference was detected (Table 2).
Response 1: In lines 77–78, there was no intention to compare or favor the use of ejaculated semen over epididymal semen. Our aim was to highlight that, based on the findings of this study, this technique can be incorporated into genetic resource preservation programs, given its applicability for recovering and preserving semen from valuable animals in cases where semen collection via ejaculation is not possible, such as post-mortem (the text has been revised to clarify this point). In lines 318–319, we discuss the superior post-thaw quality of epididymal semen reported in other studies, supporting its viability as an alternative when the use of ejaculated semen is not feasible.
Comments 2. It is unclear what the authors suggest the significance of cfDNA is in the present study.
Response 2. In the third paragraph (lines 66–74), we highlighted the relevance of cfDNA as a marker of semen quality for cryopreservation, supporting our proposal to characterize cfDNA in the semen of Pantaneiro bulls and to investigate its association with semen freezability.
Specific points:
Comments 3. Line 111: Standard error (deviation) should be indicated.
Response 3: We thank the reviewer for the valuable suggestion. As requested, we have now included the mean and standard deviation for the animals' body weight. Based on the weights of the five bulls (425, 430, 438, 446, and 451 kg), the average weight was 438 ± 9.96 kg (mean ± standard deviation). This information has been added to the manuscript (line 116)..
Comments 4. Line 145: How did it apply the pressure?
Response 4. Pressure was manually applied by the technician during testicular manipulation (this has been clarified in line 150).
Comments 5. Line 218: Is 31.1 ng/ml average of median (Figure 1)? What does “[]” mean?
Response 5: The value of 31.1 ng/µL represents the median cfDNA level among the five bulls. Although the original title referred to it as the “average,” the red dashed line corresponds to the median. Therefore, the title was corrected to identify 31.1 ng/µL as the median, and the brackets “[ ]” were removed.
Comments 6. Line 259: Was there a relation between suboptimal post-thaw sperm quality and cfDNA (cannot access Sup. File 1). Which specific individuals were excluded?
Response 6: We sincerely thank the reviewer for this important observation. We acknowledge that the supplementary material was mistakenly not included in the initial submission — we apologize for this oversight. The supplementary file is now provided for proper reference. As detailed in the supplementary table, two bulls presented post-thaw sperm motility below 20% in the ejaculated semen samples. Consequently, these individuals were not included in any IVP procedures, neither in the Ejaculate nor in the Epididymis group, in order to avoid any bias or misinterpretation in the results.
We are deeply grateful for your time and expertise, which have significantly strengthened our manuscript. Your suggestions not only improved this paper but also provided valuable directions for our future work on cfDNA and epididymal sperm cryopreservation. We hope the revised version meets your expectations and look forward to your further guidance.
Warm regards,
Reviewer 2 Report
Comments and Suggestions for Authors
This study focuses on the cryopreservation effect and free DNA (cfDNA) characteristics of Pantanero bulls, which has clear scientific significance and application value. The experimental design is reasonable and the data are detailed. The conclusions are of reference significance for the protection of genetic resources of local cattle breeds and the optimization of assisted reproductive technology.
Suggestions for improvement:
- Introduction and literature review Complementary special value: The feasibility of obtaining after the death of the bull and its irreplaceability for the preservation of genetic material of rare individuals can be further emphasized in the introduction (e.g. in conjunction with the case of endangered species conservation). Optimize literature citation: Some literature labeling is not standardized (such as [4], [21] does not fully present titles or links), it is recommended to check the format and supplement key studies (such as regional literature on the protection of native cattle breeds).
- Partial Clarify the ingredients of the cryoprotectant: "cell extender (Reprodux)" is mentioned in the article, and its specific ingredients (such as sugar, protein, antifreeze, etc.) need to be supplemented to enhance reproducibility. Refinement IVF screening: In "2.2. 7 In vitro embryo production", the specific parameters of Percoll gradient centrifugation (such as concentration, centrifugal force, time) are not clarified, and it is recommended to supplement. Sample Size Explanation: 2 bulls were excluded from the IVF experiment (due to poor thawing), and the exclusion needs to be explained in to avoid affecting the interpretation of the statistical efficacy of the results.
- Results and discussions Clarification of the expression of in vitro fertilization results: "no significant differences in cleavage and blastocyst formation" was mentioned in the discussion, but Table 2 showed that the blastocyst rate in the group was slightly higher (26.1% vs. 22.2%), and it is necessary to explain whether the small sample size led to insufficient test efficacy, or suggested supplementary efficacy (power analysis). Deepening cfDNA: The results describe only the range of cfDNA concentrations, and the discussion can compare cfDNA data from other cattle breeds (such as Nellore) (such as Dode et al. 2024) and explore their potential association with morphological defects (such as whether high cfDNA corresponds to more morphological abnormalities). Revise the wording of the conclusion: There is a contradiction between "superior... in vitro fertility" in the conclusion and "no significant difference" in the results. It is recommended to "better vitality after thawing, and equivalent in vitro fertilization ability" to ensure that the conclusion is consistent with the data.
- Format and language Uniform terminology: The full text of "cfDNA" and "cell-free DNA" needs to be expressed uniformly (it is recommended to give priority to the use of "cfDNA" and indicate the full name when it first appears). Check the chart number: the subgraph description of Figure 2 (such as A, B, C, etc.) corresponds to the actual data, but some legend annotations (such as "Epididymis (dashed line and circle)") can be clearer and avoid confusion.
Correct grammar and spelling: A few statements have repetition or redundancy (such as "Our goal was to explore the possibility... which requires assessing its mobility..." in the summary can be simplified), and it is recommended to read through the polish.
Comments on the Quality of English LanguageCorrect grammar and spelling: A few statements have repetition or redundancy (such as "Our goal was to explore the possibility... which requires assessing its mobility..." in the summary can be simplified), and it is recommended to read through the polish.
Author Response
This study focuses on the cryopreservation effect and free DNA (cfDNA) characteristics of Pantanero bulls, which has clear scientific significance and application value. The experimental design is reasonable and the data are detailed. The conclusions are of reference significance for the protection of genetic resources of local cattle breeds and the optimization of assisted reproductive technology.
Suggestions for improvement:
Comments 1. Introduction and literature review Complementary special value: The feasibility of obtaining after the death of the bull and its irreplaceability for the preservation of genetic material of rare individuals can be further emphasized in the introduction (e.g. in conjunction with the case of endangered species conservation). Optimize literature citation: Some literature labeling is not standardized (such as [4], [21] does not fully present titles or links), it is recommended to check the format and supplement key studies (such as regional literature on the protection of native cattle breeds).
Response 1: The comments were accepted and incorporated into the text (lines 61–63). The references were also corrected.
Comments 2. Partial Clarify the ingredients of the cryoprotectant: "cell extender (Reprodux)" is mentioned in the article, and its specific ingredients (such as sugar, protein, antifreeze, etc.) need to be supplemented to enhance reproducibility. Refinement IVF screening: In "2.2. 7 In vitro embryo production", the specific parameters of Percoll gradient centrifugation (such as concentration, centrifugal force, time) are not clarified, and it is recommended to supplement. Sample Size Explanation: 2 bulls were excluded from the IVF experiment (due to poor thawing), and the exclusion needs to be explained in to avoid affecting the interpretation of the statistical efficacy of the results.
Response 2: Information regarding the freezing medium used was added in lines 164–169, and the description of the sperm selection method was included in lines 212–216. Details concerning the thawing procedure of bull semen are available in the supplementary material.
Comments 3. Results and discussions Clarification of the expression of in vitro fertilization results: "no significant differences in cleavage and blastocyst formation" was mentioned in the discussion, but Table 2 showed that the blastocyst rate in the group was slightly higher (26.1% vs. 22.2%), and it is necessary to explain whether the small sample size led to insufficient test efficacy, or suggested supplementary efficacy (power analysis). Deepening cfDNA: The results describe only the range of cfDNA concentrations, and the discussion can compare cfDNA data from other cattle breeds (such as Nellore) (such as Dode et al. 2024) and explore their potential association with morphological defects (such as whether high cfDNA corresponds to more morphological abnormalities). Revise the wording of the conclusion: There is a contradiction between "superior... in vitro fertility" in the conclusion and "no significant difference" in the results. It is recommended to "better vitality after thawing, and equivalent in vitro fertilization ability" to ensure that the conclusion is consistent with the data.
Response 3: The considerations regarding the IVF results were included in the discussion section (lines 363–367), and the observations related to cfDNA are described in lines 374–380. Finally, the conclusion section was revised and adjusted accordingly.
Comments 4. Format and language Uniform terminology: The full text of "cfDNA" and "cell-free DNA" needs to be expressed uniformly (it is recommended to give priority to the use of "cfDNA" and indicate the full name when it first appears). Check the chart number: the subgraph description of Figure 2 (such as A, B, C, etc.) corresponds to the actual data, but some legend annotations (such as "Epididymis (dashed line and circle)") can be clearer and avoid confusion.
Response 4: Corrections and standardization of the cfDNA abbreviation were applied throughout the text. The legend of Figure 2 was revised, with proper identification of panels (A, B, C, D...) according to their respective treatments. Additionally, the description of epididymal and ejaculated semen representations was simplified and clarified.
Comments 5. Correct grammar and spelling: A few statements have repetition or redundancy (such as "Our goal was to explore the possibility... which requires assessing its mobility..." in the summary can be simplified), and it is recommended to read through the polish.
Response 5: We thank the reviewer for the valuable suggestion regarding the summary. The text has been revised to correct grammar issues and eliminate redundancy, as recommended. The revised version can be found in lines 15 to 25 of the manuscript.
We are deeply grateful for your time and expertise, which have significantly strengthened our manuscript. Your suggestions not only improved this paper but also provided valuable directions for our future work on cfDNA and epididymal sperm cryopreservation. We hope the revised version meets your expectations and look forward to your further guidance.
Warm regards,
Reviewer 3 Report
Comments and Suggestions for Authors
The topic of this paper is very interesting and very timely, as the possibilities of using epididymal spermatozoa to maximise the the potential to biobank and preserve particularly indigenous animal breeds is a very pressing issue in animal andrology.
I appreciate the experimental design that on one hand works with large animals, and on the other tests for in vitro fertilization.
I do have only a couple of questions:
- please, properly introduce cfDNA in the introduction. What is it and why it has become a potential marker of fertility?
- based on the results, it seems that cfDNA has rather controversial roles in the actual semen and/or sperm quality. The authors could speculate more as to why the actual sperm motility was in a negative correlation whereas the fertilisation ability was affected in a positive manner. What could be the explanation for such phenomenon?
- the paper states that PROC GLIMMIX in SAS on Demand (2023) was used for statistical analysis. Was SAS the statistical program? If so, please add the manufacturer of the program. Also, it is stated that ANOVA was used. Was it one-way or mixed model ANOVA? What post-hoc test was used for comparative analysis?
Author Response
Comments 1: Please, properly introduce cfDNA in the introduction. What is it and why it has become a potential marker of fertility?
Response 1: We appreciate the valuable feedback. As suggested, we have revised the paragraph accordingly on lines 67 and 82. Specifically, we added the following paragraph: "Cell-free DNA (cfDNA) in seminal plasma consists of small DNA fragments released during cellular processes such as apoptosis or necrosis. Recent studies have highlighted its potential as a biomarker for sperm quality and fertility, as elevated cfDNA levels may indicate cellular stress or damage. In humans and livestock, cfDNA has been correlated with sperm membrane integrity, capacitation status, and fertilization outcomes, making it a promising non-invasive marker for male fertility assessment [10,11]." This addition ensures readers understand the biological basis and significance of cfDNA in fertility research.
Comments 2:Based on the results, it seems that cfDNA has rather controversial roles in the actual semen and/or sperm quality. The authors could speculate more as to why the actual sperm motility was in a negative correlation whereas the fertilisation ability was affected in a positive manner. What could be the explanation for such phenomenon?
Response 2: The dual role of cfDNA may reflect distinct biological mechanisms. Lower cfDNA could indicate stable membranes, preserving motility, while moderate elevations might reflect active chromatin remodeling or signaling pathways that facilitate embryo development. Alternatively, cfDNA could originate from non-sperm cells (e.g., leukocytes) or reflect residual bodies from spermatogenesis, which do not impair oocyte activation. Further studies are needed to dissect these pathways and validate cfDNA's predictive value across larger cohorts. This addition aims to provoke further research while acknowledging the complexity of cfDNA dynamics.
Comments 3:The paper states that PROC GLIMMIX in SAS on Demand (2023) was used for statistical analysis. Was SAS the statistical program? If so, please add the manufacturer of the program. Also, it is stated that ANOVA was used. Was it one-way or mixed model ANOVA? What post-hoc test was used for comparative analysis?
Response 3: Thank you for highlighting this oversight. We have revised the Methods section (2.2.8) to include these details: "Data were analyzed using PROC GLIMMIX in SAS® OnDemand for Academics (SAS Institute Inc., Cary, NC, USA). A mixed-model ANOVA was employed to account for fixed effects (group, time, interaction) and random effects (replication). For post-hoc comparisons, Tukey’s HSD test was used to adjust for multiple comparisons, with significance set at P ≤ 0.05. Residuals were checked for normality and homogeneity, and log-transformed where necessary." This clarification ensures transparency and reproducibility of our statistical approach.
We are deeply grateful for your time and expertise, which have significantly strengthened our manuscript. Your suggestions not only improved this paper but also provided valuable directions for our future work on cfDNA and epididymal sperm cryopreservation. We hope the revised version meets your expectations and look forward to your further guidance.
Warm regards,
Round 2
Reviewer 1 Report
Comments and Suggestions for Authors
I have no further comment.
Reviewer 2 Report
Comments and Suggestions for Authors
They have revised this manuscript and improved significantly. Thus, i suggest to accept and publish in your journal.